# Mono- and Dimeric Sorbicillinoid Inhibitors Targeting IL-6 and IL-1β from the Mangrove-Derived Fungus *Trichoderma reesei* BGRg-3

**DOI:** 10.3390/ijms242216096

**Published:** 2023-11-08

**Authors:** Yufeng Liu, Tao Chen, Bing Sun, Qi Tan, Hui Ouyang, Bo Wang, Huijuan Yu, Zhigang She

**Affiliations:** 1School of Chemistry, Sun Yat-sen University, Guangzhou 510275, China; liuyf76@mail2.sysu.edu.cn (Y.L.); chent296@mail2.sysu.edu.cn (T.C.); sunb33@mail.sysu.edu.cn (B.S.); tanq27@mail2.sysu.edu.cn (Q.T.); ceswb@mail.sysu.edu.cn (B.W.); 2Guangdong Key Laboratory of Animal Conservation and Resource Utilization, Institute of Zoology, Guangdong Academy of Sciences, Guangzhou 510260, China; ouyh0807@gmail.com

**Keywords:** sorbicillinoids, mangrove-derived fungus, *Trichoderma reesei*, anti-inflammatory activities, IL-6, IL-1β

## Abstract

Four new sorbicillinoids, named trichodermolide E (**1**), trichosorbicillin J (**2**), bisorbicillinolide B (**3**), and demethylsorbiquinol (**5**), together with eight known compounds (**4**, **6**–**12**), were isolated from the cultures of the mangrove-derived fungus *Trichoderma reesei* BGRg-3. The structures of the new compounds were determined by analyzing their detailed spectroscopic data, while the absolute configurations were further determined through electronic circular dichroism calculations. Snatzke’s method was additionally used to determine the absolute configurations of the diol moiety in **1**. In a bioassay, compounds **7** and **10** performed greater inhibitory activities on interleukin-6 and interleukin-1β than the positive control (dexamethasone) at the concentration of 25 μM. Meanwhile, compounds **5** and **6** showed potent effects with stronger inhibition than dexamethasone on IL-1β at the same concentration.

## 1. Introduction

The mangrove is a complex ecosystem occurring in tropical and subtropical intertidal estuarine zones [1]. The special geographical location of mangroves provides a special environment with high salt, hypoxia, frequent tides, and strong solar radiation, which nourishes a diverse group of microorganisms. Among them, mangrove-derived fungi play an essential role in creating structurally unique and diverse bioactive secondary metabolites, which attract the significant attention of organic chemists and pharmacologists [1,2].

Sorbicillinoids are a family of polyketides characterized by a cyclic hexaketide nucleus with a sorbyl side chain [3]. Diverse and complex sorbicillinoids have been isolated mainly from marine fungi *Aspergillus*, *Penicillium*, *Trichoderma*, *Phialocephala*, etc. [4,5,6,7]. Sorbicillinoids can be classified into monomers, bisorbicillinoids, trisorbicillinoids, and hybrid sorbicillinoids according to their structures. The unique structures of sorbicillinoids lead to various biological activities, including cytotoxic, antibacterial, antifungal, anti-inflammatory, phytotoxic, and α-glucosidase inhibitory activities [3,8,9,10,11], which makes them one of the important lead compounds for drug development.

In our ongoing research on secondary metabolites from mangrove-derived fungi [1], four new sorbicillinoids, named trichodermolide E (**1**), trichosorbicillin J (**2**), bisorbicillinolide B (**3**), and demethylsorbiquinol (**5**), were isolated from the endophytic fungus *Trichoderma reesei* BGRg-3, which came from the root of the mangrove plant *Avicennia marina* at Nansha mangrove reserve in Guangzhou, China. Meanwhile, eight known compounds, including bisorbicillinolide (**4**) [12], sorbiquinol (**6**) [13], 13-hydroxy-trichodermolide (**7**) [6], 14-hydroxybislongiquinolide (**8**) [14], demethyltrichodimerol (**9**), trichodimerol (**10**) [15], (+)-(*R*)-vertinolide (**11**) [16], and saturnispols H (**12**) [10], were also obtained from the fungus BGRg-3 (Figure 1). The separation process, structural characteristics, and biological activities of the isolated sorbicillinoids are reported.

## 2. Results and Discussion

### 2.1. Structure Identification

Trichodermolide E (**1**) was isolated as a yellow oil. Its molecular formula was confirmed as C_18_H_22_O_7_ by using high resolution electrospray ionization mass spectroscopy (HR-ESIMS) (*m*/*z* 351.1433 [M+H]^+^, calcd. for C_18_H_23_O_7_, 351.1438, Appendix A) and NMR data, showing that eight degrees of unsaturation were required. The ^1^H NMR spectrum (Table 1) displayed four olefinic protons (*δ*_H_ 7.25 (dd, *J* = 15.6, 9.6 Hz, H-13), 6.28 (overlap, H-14, H-15), and 6.18 (d, *J* = 15.6 Hz, H-12)), two methine groups (*δ*_H_ 5.38 (dd, *J* = 3.1, 3.1 Hz, H-17) and 3.06 (t, *J* = 5.2 Hz, H-9)), two methylene groups (*δ*_H_ 3.98 (dd, *J* = 12.3, 2.8 Hz, H-18), 3.88 (dd, *J* = 12.3, 3.1 Hz, H-18), 3.61 (dd, *J* = 17.3, 5.3 Hz, H-10) and 2.86 (dd, *J* = 17.3, 5.3 Hz, H-10)), and three methyl groups (*δ*_H_ 1.86 (d, *J* = 5.0 Hz, H_3_-16), 1.85 (s, H_3_-7), and 1.42 (s, H_3_-8)). Then, 18 carbon signals in the ^13^C NMR spectrum (Table 1) were classified into three methyls (*δ*_C_ 24.2, 18.8 and 11.0), two methylenes (*δ*_C_ 62.4 and 35.2), two methines (*δ*_C_ 81.8 and 45.2), two quaternary carbons (*δ*_C_ 76.4 and 69.4), six olefinic carbons (*δ*_C_ 151.2, 144.4, 141.4, 132.2, 131.7 and 129.0), two carbonyls (*δ*_C_ 202.9, and 201.3), and one ester group (*δ*_C_ 175.4) using DEPT (distortionless enhancement by polarization transfer) spectra. Based on the HSQC (heteronuclear single quantum coherence) spectrum, the connectivity between protons and their associated carbons was confirmed (Table 1). 

^1^H-^1^H COSY (proton-proton correlation spectroscopy) correlations (Figure 2) proved that there was a long side chain (from H-9 to H-10, from H-12 to H_3_-16) and a short one (from H-17 to H-18). Based on the HMBC (heteronuclear multiple bond correlation) correlations (Figure 2) from H-12 to C-11, from H-13 to C-11, and from H-10 to C-11, a sorbyl side chain was constructed for **1**. The HMBC correlations from H_3_-7 to C-3, C-4, and C-5, from H_3_-8 to C-5, C-6, and from H-17 to C-1, C-2, C-3, and C-4 showed the presence of a heterocycle. Other HMBC correlations from H-9 to C-1, C-2, C-6, and C-8 exhibited the connection of the sorbyl side chain and the heterocycle. Thus, the constitutional formula of compound **1** was identified.

The relative configuration of compound **1** was determined using coupling constants and a NOESY (nuclear overhauser effect spectroscopy) spectrogram (Figure 3). The double bonds of the side chain were supposed to be *E* geometries according to the coupling constants (15.6 Hz for *J*_H-12/H-13_) and the NOESY correlations of H-13/H-15 and H-12/H-14. Due to the heterocycle skeleton, the configuration of C-2 and C-6 was assumed to be 2*S*, 6*S*, or 2*R*, 6*R*. Additional NOESY correlations of H_3_-8/H-9 and H_3_-7/H-9 indicated that the relative configuration of the skeleton was 2*R**, 6*R**, 9*R**.

The absolute configuration of compound **1** was confirmed based on the ECD (electronic circular dichroism) calculation and Snatzke’s method [17,18]. In order to determine the absolute configuration of the skeleton, ECD calculations were performed in B3LYP functional with the 6-311+G(d,p) basis set, using the B3LYP/6-31G(d) optimized geometries which were found through the conformer distribution calculation in the MMFF force field. Supported by the comparison of the experimental ECD spectrum with the calculated one (Figure 4a), the absolute configuration of the skeleton was confirmed to be 2*R*, 6*R*, 9*R*. Then, Snatzke’s method was used to determine the absolute configuration of acyclic 1,2-diols. Based on the Mo_2_(OAc)_4_-induced ECD measurement (Figure 4b), the 17*S* configuration was confirmed by the positive cotton effect at 360 nm [18]. Thus, the absolute configuration of compound **1** was defined as 2*R*, 6*R*, 9*R,* and 17*S*. 

Trichosorbicillin J (**2**) was expected to have a molecular formula of C_15_H_18_O_6_, confirmed through the HR-ESIMS (*m/z* 293.1034 [M−H]^−^, calcd. for C_15_H_17_O_6_, 293.1031, Appendix A), and was obtained as a yellow oil. The ^13^C NMR spectrum (Table 1) showed 15 carbons, and they were classified via the DEPT spectrum into one methyl (*δ*_C_ 7.6), three methylenes (*δ*_C_ 46.2, 36.8 and 29.3), one methine (*δ*_C_ 68.8), two olefinic carbons (*δ*_C_ 148.8, 124.3), six aromatic carbons (*δ*_C_ 164.4, 164.0, 130.8, 114.2 and 112.3), one carbonyl (*δ*_C_ 204.7), and one carboxyl (*δ*_C_ 171.8). Then, 14 protons were connected with their associated carbons according to the HSQC spectrum (Table 1). The COSY correlations (Figure 2) showed the existence of a side chain from C-8 to C-13. The COSY correlation of H-4/H-5, together with the coupling constant (8.9 Hz for *J*_H-4/H-5_), showed the ortho arrangement of them. The HMBC correlation (Figure 2) from H-12 to C-14 proved the existence of the carboxyl (C-14, *δ*_C_ 171.8). Based on the HMBC correlation from H_3_-15 to C-1, C-2, C-3, from H-4 to C-2, C-3 and from H-5 to C-1, C-6, C-7, a sorbicillinoid skeleton was built for **2**. Other HMBC correlations (from H-8 and H-9 to C-7) exhibited the connection of the side chain and the sorbicillinoid skeleton. Comparing the specific rotation with the similar compound isotrichosorbicillin E (${\left[\alpha \right]}_{D}^{20}$ = +8.1 (c 0.07, MeOH)) [19], the configuration of **2** (${\left[\alpha \right]}_{D}^{20}$ = +4.9 (c 0.1, MeOH)) was identified as 9*R*.

Bisorbicillinolide B (**3**) showed the appearance of a red oil and its molecular formula was C_28_H_36_O_8_ based on the HR-ESIMS (*m/z* 499.2346 [M−H]^−^, calcd. for C_28_H_35_O_8_, 499.2326, Appendix A). According to the ^13^C NMR spectrum (Table 2), 28 carbon signals were found and were then classified into six methyls, four methylenes, three methines, six quaternary carbons, four alkene carbons, one enol carbon, one carboxyl, and three carbonyls based on the DEPT and HSQC spectra, thus revealing a similar structure to the known compound bisorbicillinolide (**4**) [12]. Analyzing the NMR data (Table 2), in association with the indices of hydrogen deficiency, **3** was assumed to be structurally related to known compound **4** with a difference in the reduced Δ^2′^- and Δ^2″^-double bonds. The COSY correlations (Figure 2) proved the existence of two side chains from H-2′ to H-6′ and from H-2″ to H-6″. According to the HMBC correlations, together with the remaining COSY correlations (Figure 2), the constitutional formula of **3** was allowed to be confirmed.

The double bonds in the side chains were both proved to be *E* geometries according to the NOESY signals (Figure 3) of H-4′/H-6′ and H-4″/H-6″. The NOESY correlations of H_3_-15/H-(10-OH), H_3_-15/H_3_-13, H_3_-15/H-8, H_3_-15/H-11, and H-8/H-13 indicated that C-15, 10-OH, C-13, H-8, and H-11 were *syn*-oriented, partially showing that the relative configuration of **3** would be 4*R**, 8*S**, 9*S**, 10*S**, 11*R**. The NOESY correlations of H-8/H-2″, H_3_-14/H-2′, respectively determined the relative configuration of C-5 and C-7. An additional NOESY correlation of H-1/H-15 proved that the relative configuration of C-1 was 1*S**. Eventually, the relative configuration of **3** would be 1*S**, 4*R**, 5*R**, 7*S**, 8*S**, 9*S**, 10*S**, 11*R**. To determine the absolute configuration of **3**, ECD calculations were performed. The comparison between the experimental spectrum (Figure 5) with the calculated one could give the absolute configuration of compound **3** as 1*S*, 4*R*, 5*R*, 7*S*, 8*S*, 9*S*, 10*S*, 11*R*. 

Demethylsorbiquinol (**5**) showed its molecular formula as C_27_H_30_O_7_, confirmed through HR-ESIMS (*m*/*z* 465.1925 [M−H]^−^, calcd. for C_27_H_29_O_7_, 465.1908, Appendix A). Its ^13^C NMR data showed 27 carbons, including five methyls, three methines, two quaternary carbons, seven alkene carbons, six aromatic carbons, one enol carbon, and three carbonyls. A similar compound, sorbiquinol (**6**) [13], was obtained in the same fraction. A comparison of the NMR data is shown in Table 3, showing the absence of a methyl in **5**. 

Key COSY and HMBC correlations (Figure 2) were then used to prove the structure of **5**. The NOESY correlations (Figure 3) of H-2′/H-4′, H-3′/H-5′, and H-9/H_3_-11, together with the coupling constants (14.9 Hz for *J*_H-2′/H-3′_, 14.9 Hz for *J*_H-4′/H-5′_, and 15.0 Hz for *J*_H-9/H-10_), confirmed the *E* geometries of all double bonds. According to the NOESY correlations of H_3_-19/H-8 and H-1/H-8, the relative configuration was partially assumed to be 1*S**, 4*R**, 8*R**. Additional NOESY correlations of H-1/H-18 and H-7/H-9 confirmed the relative configuration to be 1*S**, 4*R**, 7*R**, 8*R**. The configuration 6*S** was determined based on the NOESY correlation of H_3_-20/H-2′. Thus, the relative configuration of **5** was assumed to be 1*S**, 4*R**, 6*S**, 7*R**, 8*R**, which was then confirmed to be its absolute configuration through ECD calculations (Figure 5).

### 2.2. Anti-Inflammatory Activities

The bioactivities of 12 sorbicillinoids (compounds **1**–**12**) were evaluated by measuring the effects of each compound on IL-6- and IL-1β-induced luciferase expression in RAW264.7 cells, testing under the concentration of 25 μM for 4 h. Their cytotoxicity was also evaluated at the same concentration level. The results are shown in Table 4. Among them, compounds **2**, **5**–**7**, and **10** exhibited inhibition against IL-6 and IL-1β. Compounds **7** and **10** presented their remarkable anti-inflammatory activities, respectively, with 47% and 67% inhibition of IL-6, 85% and 87% inhibition of IL-1β, which were even more effective than the dexamethasone. Compound **5** also showed its potent inhibition of IL-6 and IL-1β (27% and 58%, respectively), which was mildly weaker than the similar compound **6**. However, compound **5** exhibited the advantage of less cytotoxicity. Moreover, compound **2** presented a moderate inhibition of both targets with little cytotoxicity.

## 3. Materials and Methods

### 3.1. General Experimental Procedures

The NMR spectra were obtained from Bruker Avance 400 MHz and 600 MHz spectrometers (Karlsrule, Germany) at room temperature. The HR-ESIMS spectra were tested on a ThermoFisher LTQ-Or-bitrap-LC-MS spectrometer (Palo Alto, CA, USA). Optical rotation data were collected from an MCP300 (Anton Paar, Shanghai, China). UV spectra were recorded on a Shimadzu UV-2600 spectrophotometer (Shimadzu, Kyoto, Japan). The ECD experiments were processed on a J-810 spectropolarimeter (JASCO, Tokyo, Japan). Silica gel (200–300 mesh, Marine Chemical Factory, Qingdao, China) and Sephadex LH-20 (Amersham Pharmacia, Piscataway, NJ, USA) were used to perform column chromatography (CC). Semi-preparative HPLC (Ultimate 3000 BioRS, Thermo Scientific, Bremen, Germany) was used to purify the compounds.

### 3.2. Fungal Material

Fungus BGRg-3 was isolated from the root of mangrove plant *Avicennia marina*, which was collected from the Nansha Mangrove National Nature Reserve in Guangdong Province, China. The strain was then identified as *Trichoderma reesei* based on the analysis of its ITS sequence (deposited in GenBank, accession no OR353740). The fungus is now deposited at Sun Yat-sen University, China.

### 3.3. Fermentation

The fungus *Trichoderma reesei* BGRg-3 was cultivated on solid cultured medium in 1 L erlenmeyer flasks, containing 50 g of rice and 80 mL of 3‰ saline water. A total of 100 flasks were fermented for 30 days at room temperature.

### 3.4. Extraction and Purification

When the fermentation was carried out, the mycelia, as well as the rice medium, was extracted four times with MeOH and EtOAc. Eventually, a 95.5 g extract was collected (Appendix A). Then, the crude extract was eluted with a gradient elution with petroleum ether (PE) and EtOAc (from 9:1 to 0:10) on silica gel CC to obtain six fractions (Fr. A–F).

Fr.A was first fractionated on a Sephadex LH-20 column with MeOH/CH_2_Cl_2_ (1:1) to afford 2 fractions (Fr.A.1 to Fr.A.2). Fr.A.1 was then subjected to silica gel CC eluting with CH_2_Cl_2_/MeOH (100:1) to give **3** (10.3 mg). Fr.A.2 was applied to silica gel CC with CH_2_Cl_2_/MeOH (60:1) to afford a mixture of **5** (8.4 mg) and **6** (21.4 mg), which was then separated via RP-HPLC with MeOH/H_2_O (80:20). Fr.B was successively subjected to a Sephadex LH-20 column with MeOH/CH_2_Cl_2_ (1:1) and silica gel CC with CH_2_Cl_2_/MeOH (60:1) to give **4** (7.2 mg). Fr.C was also fractionated on a Sephadex LH-20 column with MeOH/CH_2_Cl_2_ (1:1) to afford 2 fractions (Fr.C.1 to Fr.C.2). Fr.C.1 was purified on RP-HPLC with MeOH/H_2_O (70:30) to afford **1** (5.3 mg). Then, **7** (6.6 mg) was obtained from Fr.C.2 by using silica gel CC eluting with CH_2_Cl_2_/MeOH (40:1). Fr.D was successively subjected to a Sephadex LH-20 column with MeOH/CH_2_Cl_2_ (1:1) and silica gel CC with CH_2_Cl_2_/MeOH (40:1) to give **9** (9.7 mg), **10** (11.2 mg), and **12** (13.5 mg). Each of them was purified via RP-HPLC with MeCN/H_2_O (75:25) and a Sephadex LH-20 column with MeOH. Fr.E was separated into Fr.E.1 and Fr.E.2 on Sephadex LH-20 column with MeOH/CH_2_Cl_2_ (1:1). Fr.E.1 was purified via RP-HPLC with MeCN/H_2_O (70:30) to give **2** (6.7 mg). Fr.E.2 was then subjected to normal-phase HPLC with *n*-hexane/2-propanol (75:25) to afford **8** (7.5 mg). Fr.F was successively subjected to Sephadex LH-20 column with MeOH/CH_2_Cl_2_ (1:1) and silica gel CC with CH_2_Cl_2_/MeOH (30:1) to give **11** (7.9 mg).

Trichodermolide E (**1**): yellow oil (5.3 mg). ${\left[\alpha \right]}_{D}^{20}$ = +18.1 (c 0.1, MeOH); UV (MeOH) *λ*_max_ (log *ε*): 238 (2.42), 271 (2.52); ECD (MeOH) *λ*_max_ (Δ*ε*): 238 (+4.12), 264 (+1.35), 342 (−0.91), see Appendix A; ^1^H NMR (CD_3_OD, 600 MHz), ^13^C NMR (CD_3_OD, 600 MHz) and other NMR data, see Table 1 and Appendix A; HR-ESIMS *m/z* 351.1433 [M+H]^+^ (calcd. for C_18_H_23_O_7_, 351.1438), see Appendix A.

Trichosorbicillin J (**2**): yellow oil (6.7 mg). ${\left[\alpha \right]}_{D}^{20}$ = +4.9 (c 0.1, MeOH); UV (MeOH) *λ*_max_ (log *ε*): 213 (2.92), 286 (2.65), see Appendix A; ^1^H NMR (CD_3_OD, 400 MHz), ^13^C NMR (CD_3_OD, 400 MHz) and other NMR data, see Table 1 and Appendix A; HR-ESIMS *m/z* 293.1034 [M−H]^−^ (calcd. for C_15_H_17_O_6_, 293.1031), see Appendix A.

Bisorbicillinolide B (**3**): red oil (10.2 mg). ${\left[\alpha \right]}_{D}^{20}$ = +203.9 (c 0.1, MeOH); UV (MeOH) *λ*_max_ (log *ε*): 306 (2.27); ECD (MeOH) *λ*_max_ (Δ*ε*): 214 (−5.03), 247 (+3.41), 286 (−1.42), 333 (+7.75), see Appendix A; ^1^H NMR (CDCl_3_, 400 MHz), ^13^C NMR (CDCl_3_, 400 MHz) and other NMR data, see Table 2 and Appendix A; HR-ESIMS *m/z* 499.2326 [M−H]^−^ (calcd. for C_28_H_35_O_8_, 499.2346), see Appendix A.

Demethylsorbiquinol (**5**): yellow oil (8.4 mg). ${\left[\alpha \right]}_{D}^{20}$ = +223.3 (c 0.1, MeOH); UV (MeOH) *λ*_max_ (log *ε*): 213 (2.62), 237 (2.42), 287 (252), 338 (2.65), 360 (2.63); ECD (MeOH) *λ*_max_ (Δ*ε*): 214 (−1.37), 239 (+4.97), 289 (−12.18), 306 (−9.47), 342 (+11.88), see Appendix A; ^1^H NMR (CDCl_3_, 400 MHz), ^13^C NMR (CDCl_3_, 400 MHz) and other NMR data, see Table 3 and Appendix A; HR-ESIMS m/z 465.1925 [M−H]^−^ (calcd. for C_27_H_29_O_7_, 465.1908), see Appendix A.

### 3.5. ECD Calculation

Conformational analyses were carried out via random searching with an MMFF force field using *Spartan’14* (v1.1.0) [20]. Subsequently, the conformers were re-optimized with density functional theory (DFT) methods at the B3LYP/6-31+G (d, p) level, solvated in methanol, with the CPCM method using the *Gaussian 09* [21]. Then, TDDFT calculations were performed at the B3LYP/6-311+G(d, p) level (nstates = 50) in methanol. The ECD spectra were obtained by overlapping Gaussian functions (σ = 0.3 eV). To obtain the final spectra, the simulated spectra of each conformer were averaged weighted with Boltzmann distribution and their relative Gibbs free energy, using the program *SpecDis* (v1.64) [22]. The theoretical ECD spectra of the corresponding enantiomers were obtained through direct inversions of the ECD spectra of the abovementioned conformers.

### 3.6. Absolute Configurations of the 17,18-Diol Moiety in ***1***

A modified version of Snatzke’s method was performed according to the published literature [17,18]. Mixtures of 1:1.2 diol-Mo_2_(OAc)_4_ for **1** were subjected to ECD measurement at a compound concentration of 1.0 mg/mL in DMSO for each. After mixing, the first ECD was recorded immediately, and the time evolution was surveyed until stationary (about 10 min after mixing). The inherent ECD was subtracted. In the induced ECD spectra, the observed signs of the diagnostic bands at around 310 and 400 nm were correlated to the absolute configuration of the 17,18-diol moiety.

### 3.7. Anti-Inflammatory Assay

The real-time PCR method was performed to measure the expression of the LPS-induced inflammatory factors IL-1β and IL-6 in RAW264.7 cells. Cells were randomly divided into control, stimulation, and administration groups, with 3 wells in each group. The control groups were cultured with a complete culture medium, the stimulation groups were stimulated with LPS (final concentration in 1 µg/mL), and the administration groups were stimulated with the mixture of LPS (final concentration in 1 µg/mL) and test compounds (final concentration in 25 µM). After culturing in an incubator for 4 h, reverse transcription was performed. Real-time PCR was performed to quantitatively analyze the products. The difference in CT values between the target gene and the internal reference gene GADPH (ΔCT) was used to perform the quantitative detection of genes in the samples. Results were given with the 2^−△△CT^ values of the same gene in different samples, which were then converted into inhibition ratios. 

## 4. Conclusions

Our work discovered four new sorbicillin-based derivatives from the secondary metabolites of *Trichoderma reesei* BGRg-3, derived from the mangrove plant *Avicennia marina*. Compound **1** exhibited a special skeleton that was rare in the sorbicillinoid family. Its 1D and 2D NMR data gave the relative configuration, and its absolute configuration was deduced based on ECD calculations and Snatzke’s method. Compound **2** was a derivative of sorbicillinol. Based on the comparison with similar compounds, constitutional formulas of compounds **3** and **5** were confirmed. Their absolute configurations were also determined through ECD calculations. In the bioactive evaluation, known compounds **7** and **10** exhibited the highest inhibition of IL-6 and IL-1β among the tested compounds, even more effective than dexamethasone. Compounds **5** and **6** also presented potent inhibition of IL-1β and moderate inhibition of IL-6. In addition, compound **6** showed the advantage of little cytotoxicity, which makes it possible to develop a nontoxic anti-inflammatory drug. Compound **8** exhibited remarkable cytotoxicity against RAW264.7 cells at the test concentration level; therefore further research on its cytotoxicity against other cells is expected.

## Figures and Tables

**Figure 1 ijms-24-16096-f001:**
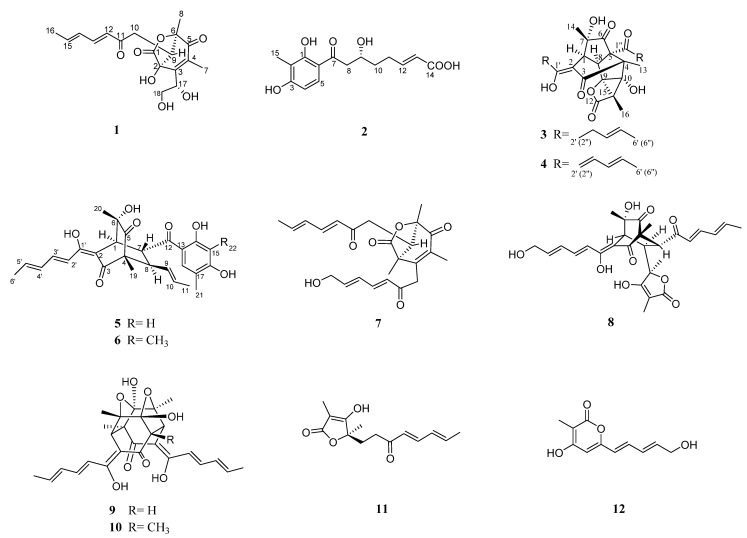
Structure of compounds **1**–**12**.

**Figure 2 ijms-24-16096-f002:**
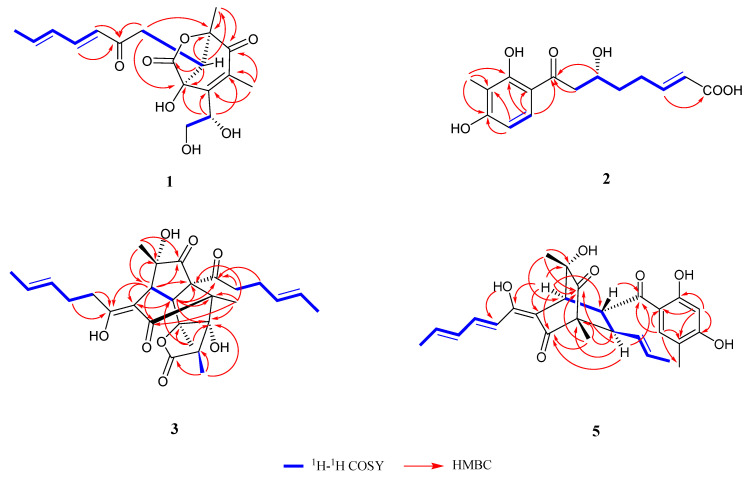
Key HMBC and COSY correlations of **1**–**3** and **5**.

**Figure 3 ijms-24-16096-f003:**
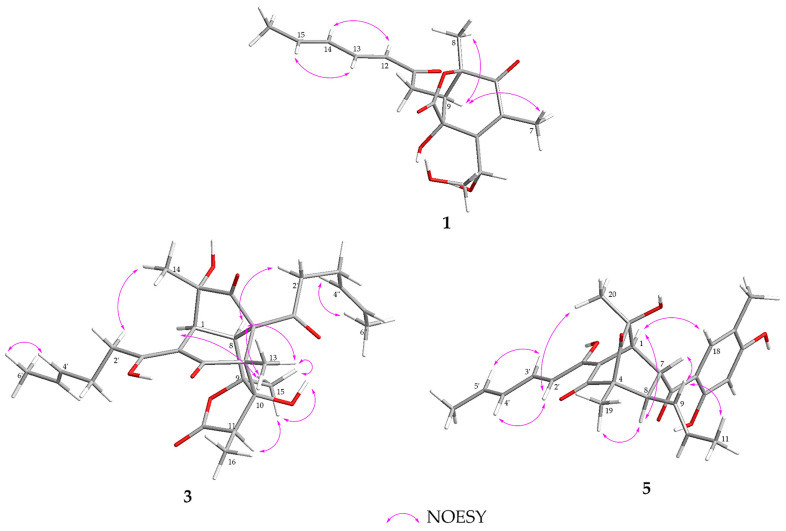
Key NOESY correlations of compounds **1**, **3**, and **5**.

**Figure 4 ijms-24-16096-f004:**
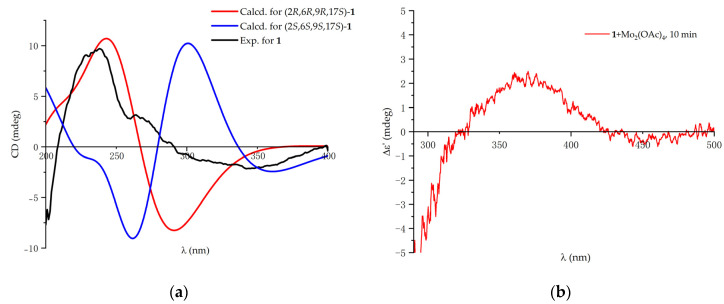
(**a**) Experimental and calculated ECD spectrum of **1**; (**b**) Mo_2_(OAc)_4_-induced ECD spectrum of **1**.

**Figure 5 ijms-24-16096-f005:**
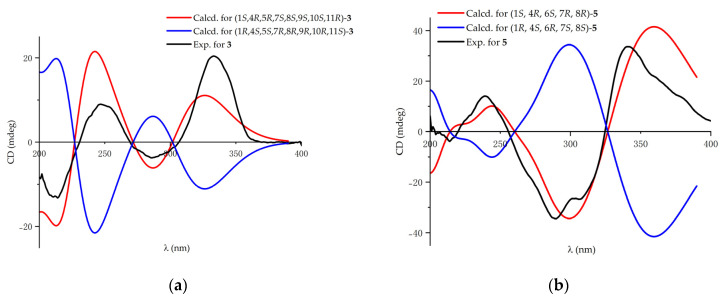
(**a**) Experimental and calculated ECD spectrum of **3**; (**b**) experimental and calculated ECD spectrum of **5**.

**Table 1 ijms-24-16096-t001:** ^1^H and ^13^C NMR data of **1** and **2** (*δ* (ppm)).

Position	1 (CD_3_OD)		2 (CD_3_OD)	
	*δ* _C_	*δ*_H_ (*J* in Hz)	*δ* _C_	*δ*_H_ (*J* in Hz)
1	175.4 (C)		164.0 (C)	
2	69.4 (C)		114.2 (C)	
3	151.2 (C)		164.4 (C)	
4	132.2 (C)		108.0 (CH)	6.4 (1H, d, *J* = 8.9)
5	202.9 (C)		130.8 (CH)	7.62 (1H, d, *J* = 8.9)
6	76.4 (C)		112.3 (C)	
7	11.0 (CH_3_)	1.85 (3H, s)	204.7 (C)	
8	24.2 (CH_3_)	1.42 (3H, s)	46.2 (CH_2_)	3.13 (1H, dd, *J* = 15.6, 8.0)3.01 (1H, dd, *J* = 15.6, 4.4)
9	45.2 (CH)	3.06 (1H, t, *J* = 5.2)	68.8 (CH)	4.2 (1H, tt, *J* = 8.4, 4.4)
10	35.2 (CH_2_)	3.61 (1H, dd, *J* = 17.3, 5.2)2.86 (1H, dd, *J* = 17.3, 5.2)	36.8 (CH_2_)	1.69 (2H, m)
11	201.3 (C)		29.3 (CH_2_)	2.36 (2H, m)
12	129.0 (CH)	6.18 (1H, d, *J* = 15.6)	148.8 (CH)	6.92 (1H, dt, *J* = 15.4, 6.9)
13	144.4 (CH)	7.25 (1H, dd, *J* = 15.6, 9.6)	124.3 (CH)	5.87 (1H, d, *J* = 15.4)
14	131.7 (CH)	6.28 (overlap)	171.8 (C)	
15	141.4 (CH)	6.28 (overlap)	7.6 (CH_3_)	2.04 (3H, s)
16	18.8 (CH_3_)	1.86 (3H, d, *J* = 5.0)		
17	81.8 (CH)	5.38 (1H, dd, *J* = 3.1, 3.1)		
18	62.4 (CH_2_)	3.98 (1H, dd, *J* = 12.3, 3.1)3.88 (1H, dd, *J* = 12.3, 3.1)		

**Table 2 ijms-24-16096-t002:** ^1^H and ^13^C NMR data of **3**–**4** (*δ* (ppm)).

Position	3 (CDCl_3_)		4 (CDCl_3_)	
	*δ* _C_	*δ*_H_ (*J* in Hz)	*δ* _C_	*δ*_H_ (*J* in Hz)
1	44.2 (CH)	3.25 (1H, d, *J* = 5.1)	44.4 (CH)	3.38 (1H, d, *J* = 5.4)
2	108.0 (C)		107.3 (C)	
3	185.3 (C)		194.5 (C)	
4	65.3 (C)		68.0 (C)	
5	73.5 (C)		73.5 (C)	
6	210.2 (C)		210.3 (C)	
7	84.2 (C)		84.1 (C)	
8	56.5 (CH)	3.62 (1H, d, *J* = 5.1)	56.9 (CH)	3.73 (1H, d, *J* = 5.4)
9	94.2 (C)		93.8 (C)	
10	88.5 (C)		88.3 (C)	
11	41.2 (CH)	2.68 (1H, q, *J* = 7.7)	41.1 (CH)	2.66 (1H, q, *J* = 8.0)
12	178.5 (C)		177.9 (C)	
13	10.0 (CH_3_)	1.24 (3H, s)	10.3 (CH_3_)	1.36 (3H, s)
14	17.4 (CH_3_)	1.23 (3H, s)	18.1 (CH_3_)	1.23 (3H, s)
15	24.2 (CH_3_)	1.32 (3H, s)	24.3 (CH_3_)	1.39 (3H, s)
16	13.8 (CH_3_)	1.19 (3H, d, *J* = 7.7)	13.7 (CH_3_)	1.21 (3H, d, *J* = 8.0)
1′	195.2 (C)		177.3 (C)	
2′	36.4 (CH_2_)	2.50, 2.32 (2H, m)	119.0 (CH)	6.14 (1H, d, *J* = 14.8)
3′	26.8 (CH_2_)	2.28 (2H, m)	144.6 (CH)	7.39 (1H, dd, *J* = 14.8, 10.4)
4′	129.4 (CH)	5.37–5.52 (m)	131.0 (CH)	6.1–6.5 (m)
5′	126.5 (CH)	5.37–5.52 (m)	141.5 (CH)	6.1–6.5 (m)
6′	18.0 (CH_3_)	1.61 (3H, d, *J* = 6.3)	18.9 (CH_3_)	1.90 (3H, d, *J* = 5.2)
1″	212.2 (C)		199.4 (C)	
2″	40.9 (CH_2_)	2.89, 2.55 (2H, m)	123.8 (CH)	6.46 (1H, d, *J* = 14.8)
3″	26.2 (CH_2_)	2.19 (2H, m)	147.9 (CH)	7.42 (1H, dd, *J* = 14.8, 11.2)
4″	128.7 (CH)	5.28 (1H, m)	130.3 (CH)	6.21 (1H, dd, *J* = 15.2, 11.2)
5″	127.1 (CH)	5.37–5.52 (m)	145.0 (CH)	6.36 (1H, dq, *J* = 15.2, 7.6)
6″	18.0 (CH_3_)	1.59 (3H, d, *J* = 6.4)	19.0 (CH_3_)	1.89 (3H, d, *J* = 7.6)

**Table 3 ijms-24-16096-t003:** ^1^H and ^13^C NMR data of **5**–**6** (*δ* (ppm)).

Position	5 (CDCl_3_)		6 (CDCl_3_)	
	*δ* _C_	*δ*_H_ (*J* in Hz)	*δ* _C_	*δ*_H_ (*J* in Hz)
1	47.1 (CH)	3.31 (1H, d, *J* = 1.8)	47.0 (CH)	3.32 (1H, d, *J* = 1.5)
2	106.7 (C)		106.8 (C)	
3	198.0 (C)		198.0 (C)	
4	63.2 (C)		63.2 (C)	
5	211.5 (C)		211.5 (C)	
6	75.9 (C)		75.7 (C)	
7	46.7 (CH)	4.27 (1H, dd, *J* = 6.5, 1.8)	46.7 (CH)	4.28 (1H, dd, *J* = 6.5, 1.5)
8	46.4 (CH)	3.23 (1H, dd, *J* = 10.0, 6.5)	46.5 (CH)	3.26 (1H, dd, *J* = 10.0, 6.5)
9	128.2 (CH)	5.02 (1H, ddq, *J* = 15.0, 10.0, 1.8)	128.6 (CH)	5.02 (1H, ddq, *J* = 15.0, 10.0, 1.5)
10	130.8 (CH)	5.44 (1H, dq, *J* = 15.0, 6.5)	130.6 (CH)	5.45 (1H, dq, *J* = 15.0, 7.0)
11	17.9 (CH_3_)	1.61 (3H, dd, *J* = 6.5, 1.8)	17.8 (CH_3_)	1.60 (3H, dd, *J* = 6.5, 1.5)
12	202.0 (C)		202.1 (C)	
13	112.8 (C)		112.0 (C)	
14	164.2 (C)		161.9 (C)	
15	103.5 (CH)	6.34 (1H,s)	110.7 (C)	
16	161.2 (C)		159.0 (C)	
17	116.4 (C)		115.0 (C)	
18	132.4 (CH)	7.70 (1H, s)	129.0 (CH)	7.59 (1H, s)
19	10.1 (CH_3_)	1.16 (3H, s)	10.0 (CH_3_)	1.16 (3H, s)
20	24.5 (CH_3_)	1.21 (3H, s)	24.3 (CH_3_)	1.20 (3H, s)
21	15.6 (CH_3_)	2.25 (3H,s)	16.0 (CH_3_)	2.26 (3H,s)
22	-	-	7.5 (CH_3_)	2.12 (3H,s)
1′	169.0 (C)		168.8 (C)	
2′	117.2 (CH)	5.57 (1H, d, *J* = 14.9)	117.2 (CH)	5.53 (1H, d, *J* = 15.0)
3′	142.8 (CH)	7.19 (1H,dd, *J* = 14.9, 10.7)	142.4 (CH)	7.17 (1H,dd, *J* = 15.0, 10.5)
4′	130.9 (CH)	5.97 (1H, m)	130.8 (CH)	5.93 (1H, ddq, *J* = 15.0, 10.0, 1.5)
5′	139.7 (CH)	6.08 (1H, dq, *J* = 14.9, 6.9)	139.4 (CH)	6.08 (1H, dq, *J* = 15.0, 7.0)
6′	18.9 (CH_3_)	1.82 (3H, d, *J* = 6.9)	18.9 (CH_3_)	1.82 (3H, dd, *J* = 7.0, 1.5)

**Table 4 ijms-24-16096-t004:** The cytotoxicity and the inhibitory effects on IL-6 and IL-1β of the tested compounds.

Raw 264.7	IL-6 Inhibition (%) ^1^	IL-1β Inhibition (%) ^1^	Viability of Cell (%) ^1^
Compound	25 μM		
**1**	NA	NA	79.85 ± 2.07
**2**	45.55 ± 9.22	21.63 ± 5.78	96.40 ± 6.26
**3**	NA	NA	98.39 ± 4.72
**4**	NA	NA	97.88 ± 2.84
**5**	26.76 ± 3.44	57.77 ±14.38	106.09 ± 0.99
**6**	34.86 ± 5.55	75.08 ± 6.80	86.72 ± 2.83
**7**	46.68 ± 11.80	84.80 ± 4.49	92.05 ± 1.86
**8**	NA	NA	55.44 ± 4.23
**9**	NA	NA	83.02 ± 0.59
**10**	66.64 ± 11.36	87.05 ± 1.94	117.24 ± 4.72
**11**	NA	NA	83.96 ± 0.70
**12**	NA	NA	84.94 ± 1.81
DEX ^2^	36.19 ± 3.75	48.19 ± 9.50	NT

^1^ Data are presented as the mean ± SD. ^2^ Positive control; NA: no activity (inhibition ≤ 20%); NT: not tested.

## Data Availability

Data are contained within the article and Appendix A.

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
