# Peer review of "Mono- and Dimeric Sorbicillinoid Inhibitors Targeting IL-6 and IL-1β from the Mangrove-Derived Fungus Trichoderma reesei BGRg-3"

_ijms, 2023, doi:10.3390/ijms242216096_

Round 1

Reviewer 1 Report (Previous Reviewer 4)

Comments and Suggestions for Authors

Dear authors,

Although the authors did not try to modify the structural elucidation as I suggested (in my opinion it is only a list of spectroscopic data without explanation), the queries have been corrected. The paper is ready for acceptance.

Author Response

Reviewer 2 Report (Previous Reviewer 3)

Comments and Suggestions for Authors

While the authors fixed most of the problems, there are still lots of clerical or grammatical errors, please carefully do a proof checking.

1. There are still lots of clerical or grammatical errors, e.g. “compound 7 and 10” should be compounds 7 and 10”; All the compounds numbers should be bolded; “which even more effective than the dexamethasone” should be “which were even more effective than dexamethasone”; “sorbicillinold family” should be “sorbicillinoid family”. Please carefully check the whole context.

Comments on the Quality of English Language

1. There are still lots of clerical or grammatical errors, e.g. “compound and 10” should be compounds and 10”; All the compounds numbers should be bolded; “which even more effective than the dexamethasone” should be “which were even more effective than dexamethasone”; “sorbicillinold family” should be “sorbicillinoid family”. Please carefully check the whole context.

Author Response

Reviewer 3 Report (Previous Reviewer 2)

Comments and Suggestions for Authors

The manuscript “Mono‐ and Dimeric Sorbicillinoid Inhibitors Targeting the IL‐6 and IL‐1β from the Mangrove‐Derived Fungus Trichoderma reesei BGRg‐3“ [now ijms-2705268] written by Yufeng Liu, Tao Chen, Bing Sun, Qi Tan, Hui Ouyang, Bo Wang, Huijuan Yu and Zhigang She has been revised and newly submitted by the authors. The reviewer is grateful to the authors for taking to heart the previous comments, also from other reviewers and for addressing several concerns.  

A list with answers of the authors to all reviewer’s comments, point by point, is added to the new submission. The authors have taken all of the comments of the reviewer as well as obviously of other the reviewers into account. The authors have answered to the concerns and made lots of respective corrections, additions, and changes in the newly submitted manuscript. The reviewer has not found any comment, which was not considered in the revised manuscript. All changes in the manuscript make sense. They have been carried out with direct reference to the various comments.

Hence, the equality and readability of the manuscript have been improved in the newly submitted version. The manuscript now fully and intelligibly describes the isolation and structural identification of four new and eight known sorbicillinoids from a Trichoderma reesei strain as well as the tests of anti inflammatory activities of these compounds. The manuscript is therefore acceptable for publication in the “International Journal of Molecular Science”.

Author Response

Reviewer 4 Report (Previous Reviewer 1)

Comments and Suggestions for Authors

The authors give positive responses for all queries raising from reviewers. I recommended that this revision is acceptable for publication with its present form. 

Author Response

This manuscript is a resubmission of an earlier submission. The following is a list of the peer review reports and author responses from that submission.

Round 1

Reviewer 1 Report

Comments and Suggestions for Authors

A paper entitled “Mono- and Dimeric Sorbicillinoid Inhibitors Targeting IL-6 and IL-1β from the Mangrove-Derived Fungus Trichoderma reesei BGRg-3” is submitted to Int. J. Mol. Sci. for further reviewing and publication. In this paper, the authors described four new sorbicillinoids and eight know compounds from the cultures of the mangrove-derived fungs. T. reesei. The structures for the new metabolites were established by spectroscopic methods and their biological activity were evaluated. I recommend that this paper is acceptable for publication after revision.

Major comments:

1.     Too many typing and grammatical errors were found. For example, in the Title “…Mangrove -Derived…” should be revised as “…Mangrove-Derived…”. The authors have to recheck the text throughout carefully.

2.     In Figure 2, in general, 4J-HMBC correlations were not recognized. Please rechecked the HMBC spectrum for all isolates.

3.     How to define the conjugated diene in 1 are existed in s-trans or s-cis form?

4.     In Figure 3, if it possible, please redraw the NOESY correlations by CS Chem3D Model. It is very difficult to understand the correlations in this Figure. For example, in compound 3, H-1 correlated with the methyl protons attached at C-7. It will be confused.

5.     In Figure 4, the curve for Calcd for (2S,6S,9S)-1 should be provided. (Figure 5 also should be provided).

6.     Page 5, line 116: isotrichosorbicillin [α]D +8.1. The optical rotation value ocompound 2 is +4.9. The rotation data are near zero. It is difficult to distinguish if 2 or its enantiomer. More evidence is necessary. If there is enough material was obtained. Mosher’s method is a optional way.

7.     In Table 3, the chemical shift for C-3 in 5 are δC 198.0; and for C-3 in 6 are δC 168.8. It is impossible. Please recheck the NMR data for these two compounds.

8.     In Table 3, the chemical shift for C-1' in 5 are δC 169.0; and for C-' in 6 are δC 198.0. It is impossible. Please recheck the NMR data for these two compounds.

9.     Following the data shown in Table 4, why the authors think compound 11 is active (page 9, line 165)? (inhibition rate = 9.33 and 10.93%).

10.  The weight for all isolates should be provided.  

Minor comments:

11.  The phrase “secondary metabolites” is not good to be a keyword. The name for the target organism “Trichoderma reesei” is an option for to be key word.

12.  The reference format “[1, 2]””[1,2]”. Please check it throughout the text.

13.  Line 36, “…so on[4-7]…””…so on [4-7]…”.

14.  In Table 1. The title “….1 and 2 (δ (ppm), J (Hz)).” “….1 and 2 (δ (ppm)).”. In the row δH δH (J in Hz). Please check all Tables.

15.  Page 6. What happen in this page, only two lines.

16.  Page 7, line 133, “bones””bonds”.

17.  In reference, is the issue number necessary?

18.  In ref. 15 and 18. Are the names for the authors capital?

In ref. 17, What is Mar. Drugs 2023, 21, (8). It is not available. Please recheck the format carefully again. 

Comments on the Quality of English Language

Extensive editing of English language required

Reviewer 2 Report

Comments and Suggestions for Authors

The manuscript “Mono‐ and Dimeric Sorbicillinoid Inhibitors Targeting the IL‐6 and IL‐1β from the Mangrove‐Derived Fungus Trichoderma reesei BGRg‐3“ [ijms-2577151] written by Yufeng Liu, Tao Chen, Bing Sun, Qi Tan, Hui Ouyang, Bo Wang, Huijuan Yu andZhigang She describes the isolation and structural identification of four new and eight known sorbicillinoids from a Trichoderma reesei strain. In particular 1D and 2D NMR, MS and optical rotation analysis combined with in silico studies have been used to determine the molecular structures. Possible anti inflammatory activities of these compounds is tested and briefly discussed.

The reviewer has expertise in the molecular field of organic chemistry and structure determination (in particular NMR) and hence mostly refers to this part of the manuscript with the review.

All structural investigations and anti inflammatory tests are performed with modern and common state of the art methods. The overall practical work is well planned and performed. The practical investigation has been made carefully with state of the art methods. The further analysis and interpretation of the data as well as the discussion and conclusion are sensible and detailed with respect to data gained. Some concerns about the presentation and a few comments about possible biosynthesis as well as further bioactivities are given in the comments.

Hence, the results possess importance in furthering our knowledge of sorbicillinoids from a Trichoderma reesei. The manuscript is therefor of some interest in the fields of Mycology, Organic Chemistry, Natural Product Chemistry and to some extent of Medicinal Chemistry and Pharmacy. The results are worth publishing in general. However, there are some points listed below, which should be further addressed by the authors. Hence, there are some comments listed below, which should be taken into account by the authors prior to acceptance of the manuscript. Therefore, the manuscript is not yet in completely a form to be published in “International Journal of Molecular Science“.

General Comments:

a) The authors describe an unexpectedly high anti-inflammatory activity of some compounds. Authors are encouraged to elaborate on why exactly this bioactivity was tested. Is there any prior evidence that the compounds or extracts of this fungus possess such properties?

b) Authors are encouraged to explore and describe (if possible) or at least discuss in the discussion, other potential bioactivities to demonstrate the importance of further investigation of the compound class and fungus.

c) The molecular structures of the four new compounds are partly similar to those of already known sorbicillinoids. The authors are strongly encouraged to discuss possible biosynthetic pathways for the new four structures in order to be able to better classify them in the substance class and their generally discussed biosynthesis.

Minor comments (scientific):

d)
Line 68: The 13C NMR signal at 175.4 ppm does not belong to a carbonyl group but to a carboxylic group.

e) Table 1, compound 1, delta H of position 17: This is not an “t”, but a “dd” (The coupling protons are diastereotopic to each other.)

f)
The reviewer always has some problems with the formulation "planar structure" (eg line 78 and elsewhere in the manuscript), since she/he always imagines a flat molecule. She/He would recommend authors to use the word "constitutional formula".

g) Chapter 3.4: The extinction coefficients for the individual UV measurements are indicated here, but no numerical values are given for them.
Authors are encouraged to report these values as well.

h) Supp. Mat.: Figures S35 and S36: Were these measurements really made in deuterated methanol, while the NMR were recorded in deuterated chloroform?

i) Figures S10, S13 and 29: Here the phase correction has not been made. Authors are encouraged to revise this. Otherwise, this reduces the quality of the very good structure elucidation a little bit.

j) With the NOESY spectra, a baseline correction could possibly increase the informative value of the spectra a little.

k) A graphical presentation of the isolation procedure (flowchart) in Supp. Mat. would made the isolation clearer.

Minor comments (presentation):

- Lines 14 and 15: Do not capitalize the names of the compounds in the sentence.
- Line 19: “… spectra. The Snatzke’s …” (with fullstop)
- Line 36 “… and so on.” Does not sound very scientific and should be rephrased.
- Line 42 (and several other positions): “… fungi [1, …]” (
space before the parenthesis)
- Line 86 “… 6R. Additional …” (with fullstop)
- Table 3, position 10: space behind the comma between coupling constants.
- Line 156 “double bonds” (not “bones”)
- line 163 please use “h” as unit instead of “hours”.
- line 178 “… Karlsruhe …” (not “karlsrule”)
- line 182 “… Japan). Silica …” (with capital “S”)
- line 191 “… OR353740). The …” (with capital “T”)
- line 199 (and other positions) “… “95.5 g” …” (blanc between number und unit)
- line 212 “… 10 and 12 …” (with blanc after 10)
- line 242 “1” in bold
- References 15 and 18: Do not write the entire names with capitals and adopt these to the other references.
- Supp. Mat. (several positions): “H,H-COSY” (without blanc)
- Supp. Mat., page 4: 1H with superscript “1” and CD3OD with subscript “3
”.

Reviewer 3 Report

Comments and Suggestions for Authors

In brief summary, the manuscript reported four new sorbicillinoids along with eight knowns from the mangrove-derived fungus Trichodermareesei BGRg-3, and evaluated their inhibitory activity against IL-6 and IL-1β. The new structures were elucidated based on the spectroscopic data, ECD, and Snatzke’s method; however, the structures are questionable. The writing also needs to be improved.

1.     For the relative configuration of compound 1, why do you assume C-2 and C-6 to be co-facial? Also, for H-9, both R and S are very close to H3-8, how do you determine the relative configuration?

2.     Compound 2, the coupling constant for H-2 and H-5 is 8.9, which seems abnormal. Please double-check

3.     Compound 3, “Analyzing the NMR data (Table 2), compound 3 owned four more methylenes (δC 40.9, 36.4, 26.8 and 124 26.2), while four alkene carbon signals (δC 147.9, 144.6, 123.8 and 119.0) were missed.” confusing sentences. Please re-write it.

4.     Page 7, the paragraph for configuration of compound 3, please make it more in detail and double-check.

A) “The NOESY correlations of H3-(7-CH3)/H-2’, H3-(7-CH3)/H-1, H-1/H-2’ and H-8/H-2’’, were found to determine the relative configuration of C-1, C-5 and C-7.” What does it mean?

B) H3-7 and H-1 showed NOE correlation, however, in Figure 3, they occupy the opposite direction;  

C) “Eventually, the relative configuration of 3 would be 1S*, 4R*, 5R*, 7S*, 8S*, 9S*, 10S*, 11R*.”, how do you conclude that?

D) Similar questions for compound 5.

5.     Please list all the activities of the tested compounds, not selected compounds.

6.     Please add the amount of the isolated compounds in part 3.4.

7.      “compound 7 and 10”  should be “compounds 7 and 10”; please check the whole text.

Comments on the Quality of English Language

Need to be improved.

Reviewer 4 Report

Comments and Suggestions for Authors

Dear authors,

The article is a classical work in isolation and structure elucidation of natural products where four new compounds of the sorbicillinoid family have been isolated and characterized.

Moreover, the article is well-written and the structures have been established through extensive spectroscopic techniques including those specifically thecniques required for the determination of the absolute stereochemistry.

Regarding the strutural elucidation I have a critical point because authors, in this paper, did not really provide the structural elucidation but they only write a sequence of spectroscopic data that leads to the establishment of the planar structure but lacks a comprehensive explanation. I know that the structure of compound 1 is closely related to other already published compounds and that gave an easy way for the elucidation of these new ones. However, in the relative configuration of compound 1, I have reservations about the configuration at C-9. The authors have established and R configuration according to the NOESY correlation observed between H8/H9. My question is whether, in the case of an opposite configuration, wouldn´t the same NOESY correlation have been observed?.

Furthermore, I would like to draw attention to the anti-inflammatory activity:

-It is essential to provide the experimental data in the “Materials and Methods” section.

-Which was the concentration of dexamethasone used as positive control? 25 micromolar?. Please include this information in the experimental section.